# Lifestyle Pathways Affecting Children’s Mental Health in Japan during the COVID-19 Pandemic

**DOI:** 10.3390/children10060943

**Published:** 2023-05-26

**Authors:** Satomi Sawa, Akihito Hagihara

**Affiliations:** 1Graduate School of Teacher Training Development, University of Toyama, 3190 Gofuku, Toyama 930-8555, Japan; 2Department of Preventive Medicine and Epidemiology, National Cerebral and Cardiovascular Center, 6-1 Kishibe Shinmachi, Suita, Osaka 564-8565, Japan

**Keywords:** COVID-19 pandemic, lifestyle, mental health, preschooler, elementary school children

## Abstract

The recent prolonged COVID-19 pandemic has worsened the daily lives of preschoolers and elementary school children worldwide. Although these changes may have affected their mental health, the full picture still remains unknown. Since March 2020, Japan has intermittently experienced several COVID-19 waves. This survey was conducted between February and March 2022. In this study, we investigated the pathways by which specific lifestyle factors (such as exercise, sleep, diet, and life skills) affect physical/psychosocial health (PPH) in 1183 preschoolers (3–5 years old) and 3156 elementary school children (6–11 years old) in Toyama Prefecture, Japan. These pathways were examined using a path analysis. Consequently, “life skills” was found to be the factor most strongly associated with PPH in both preschoolers and elementary school children (*p* < 0.001). Furthermore, it was associated with the physical activity score and with PPH via physical activity. Moreover, both boys’ and girls’ mental health declined with age among elementary school children (*p* < 0.001). The results of the current study may be helpful for early interventions (around the start of elementary school) at home and at school to improve children’s mental health during the COVID-19 pandemic.

## 1. Introduction

Numerous studies in the past few years have reported an increase in the cases of unfavorable lifestyle, mental stress, and depressive symptoms in children as a result of the prolonged COVID-19 pandemic [1,2,3]. In terms of its impact on children’s lifestyle, research has so far revealed findings on decreased physical activity [4], a postponed bed and wake-up time [5], and unhealthy eating habits [6]. The percentage of kids who spend more time on screens and lead sedentary lifestyles significantly increased during the lockdown [7]. Due to impaired circadian rhythms brought on by lockdown and travel restrictions, children experienced sleep disorders, leading to mental disorders through reduced sleep duration and postponed bed and wake-up times [8]. Both irregular mealtimes [9] and skewed nutritional balance in children were also reported [6]. Contrarily, children who regularly engaged in physical activity and played outdoors felt better physically and mentally than those who did not [10]. Furthermore, balanced nutrition and moderate physical exercise lessened children’s anxiety and depressive symptoms [11].

Children’s life skills are nurtured during their childhood through participation in various activities at home and at school [12,13]. Individual health behaviors, such as physical activity, sleep, and diet, are shown to be related to mental health in children [14,15]. However, it remains unknown how these factors have affected preschoolers’ or elementary school students’ mental health during the pandemic. The impact of COVID-19 on children’s mental health is greater than that on adults because children are unable to adequately explain and manage their mental stress [8]. It is essential to assess children’s mental health during the COVID-19 pandemic and implement a successful response.

It has been discovered that the pandemic’s impact on stress symptoms, such as anxiety, depression, and self-control, varies by age and gender, with females being the most affected [1]. Thus, when analyzing the impact of COVID-19 on children, it is important to consider the participants’ age and gender. Individual health behaviors, such as physical activity, sleep, and diet, interact with one another and may have an impact on mental health. However, findings on the process during the pandemic have not been obtained. Therefore, the present study’s aim is to describe the relationship between the lifestyle factors and mental health of preschoolers and elementary school children during the COVID-19 pandemic by gender and age.

## 2. Materials and Methods

### 2.1. Participants and Survey Outline

The design of this online study was approved by the ethics committee of the University of Toyama (Approval Code: R2021151, Approval Date: 7 February 2022). The study participants were preschoolers (aged 3–5 years) and elementary school children (aged 6–11 years) attending schools in Toyama Prefecture, Japan. The survey was conducted between February and March 2022. Since March 2020, Japan has intermittently experienced several COVID-19 waves. The first wave hit the country in early February 2020, during which, the total number of infected cases exceeded 100,000. The number of positive cases reached the record high in Toyama prefecture (approx. 3000) at the time of the survey. 

The school board and the school principal received a complete explanation of the study’s design and data management protocol prior to the survey. With the permission of their principals, a letter requesting participation in the survey was distributed to 77 elementary schools and 90 early childhood centers for preschoolers. The letter requesting participation in the study was distributed by the class teachers, and the children and their parents received a complete explanation of the study’s design and data management procedures. All the children responded to the survey questions put up on a website while they were at home with their families, which indicated the assistance of their parents in answering the questions. The participants’ informed consent was implied by their entry into the designated website. In total, 1460 preschoolers and 3747 elementary school children participated in our survey and completed the questionnaires.

### 2.2. Questionnaire

The study participants were required to respond to questionnaires about their sex, grade (ages 3–11), lifestyle, and mental health. All responses were submitted to the website. In total, 1183 (81.0%) preschoolers and 3156 (84.2%) elementary school children out of 1460 preschoolers and 3747 elementary school children who returned their completed questionnaires had provided complete answers to all pertinent questions, and their data were included in the analysis. We used questionnaire items in this study that addressed children’s lifestyle and mental health. We specifically focused on factors that have been linked to mental health in prior research [16] and eventually chose 21 individual lifestyle factors that can be changed through home and school-based health education. Prior to the path analysis, a factor analysis was performed using data from 3156 elementary school children and 1183 preschoolers. The results of the analysis revealed that there were 21 items in the factor structure, which was composed of five factors: “sleeping habits”, “physical/psychosocial health (PPH)”, “physical activity”, “life skills”, and “healthy diet” (Appendix A). Children in preschool and elementary school shared the same factor structure. There were four items totaled for the total sleeping habits score, seven items for the PPH score, four for the total physical activity score, four for the total life skills score, and two for the healthy diet score.

### 2.3. Statistical Analyses

The theoretical model was developed using theoretical arguments for children’s lifestyle, exercise, and PPH [13,14], and using previous structural and complex analysis of pathways [15,17]. Staying active during a pandemic has been reported to enhance immune system function [18] and promote positive changes in the ability to cope with stress [19]. Furthermore, a study also found that physical activity participation is linked to an improved mental health [15]. However, during the lockdown period, the proportion of children with increased sedentary lifestyles increased [7] with a subsequent decrease in physical activity [4]. Therefore, in this study, a theoretical model was prepared to reflect the situation where physical activity is no longer readily available due to the prolonged COVID-19 pandemic. Specifically, it is theoretically possible that improved mental health leads to improved physical health in the process. However, the population was urged to stay home to minimize human contact during the COVID-19 pandemic in Japan. A model treating physical health as an exogenous variable leading to physical health did not reflect an actual situation. The theoretical model of this study showed that sleep habits, life skills, and healthy diets may affect PPH directly, as well as indirectly, through physical activity.

Separate analyses were conducted for children in preschool and elementary school due to the differences in the educational curricula and growth stages between the two age groups [13,20]. The means, standard deviations, and cross-correlations of the variables in the structural equation model were calculated, as well as the Pearson’s correlation coefficient between variables. The participants were split into two subgroups: preschoolers and elementary school children, for the purpose of path analysis. 

For the structural equation model, the goodness of fit indices, the normed fit index (NFI), the goodness of fit index (GFI), the comparative fit index (CFI), and the root-mean-square error of approximation (RMSEA), were used. SPSS (ver. 28.0 J; SPSS Inc., IBM, Armonk, NY, USA) was used to conduct these analyses. The SPSS-compatible Amos software (ver. 28.0 J; SPSS Inc.) was used to conduct the path analysis. For all tests, two-tailed *p*-values under 0.05 were regarded as statistically significant.

## 3. Results

Descriptive statistics for the variables in the structural equation model for the two groups of children are shown in Table 1. Boys and girls made up the same proportion of each group. In general, the elementary school children’s overall life skills scores were >1 higher than those of the preschoolers [11.93 (2.69) vs. 10.87 (2.61)]. However, the elementary school children’s total PPH scores were typically lower than those of the preschoolers [21.43 (3.88) vs. 22.16 (3.53)]. The preschoolers and elementary school students had mean ages of 4.00 (±0.80) and 8.39 (±1.68), respectively. All of the Cronbach’s alpha reliability values for the total sleeping habits, total PPH score, physical activity score, life skills score, and healthy diet score were over 0.70 in preschoolers and elementary school students, indicating a high reliability of these scales.

The correlation coefficients of the variables used in the path analyses are broken down by type of school in Table 2. The correlation coefficients between the variables in the two groups are displayed by the lower and upper diagonals. Age and sex were related to the total life skills score in the preschoolers’ upper diagonals (*p* < 0.001). Five types of total scores were correlated with one another (*p* < 0.001). Age and sex were unrelated in the elementary school children’s lower diagonal. Similarly to the situation with preschoolers, five types of total scores were correlated with one another (*p* < 0.001).

First, a theoretical model (Figure 1) that considered the roles that a school and a family play in a child’s health education was used as a basis for path analysis. Second, in each of the two groups, non-significant paths (*p* ≥ 0.05) were removed, and a modified model was created. The fitness indices for the initial and revised models are shown in Table 3 by the types of children (i.e., preschoolers and elementary school children). The revised path models for the two groups are displayed in Figure 2 and Figure 3. All values represent standardized beta coefficients. The revised models showed a good fit, as shown in Table 3, with nonsignificant chi-square values for preschoolers (χ^2^ = 11.896 (df = 9), *p* > 0.05) and elementary school children (χ^2^ = 4.423 (df = 2), *p* > 0.05). Additionally, the *p* values for the RMSEA were less than 0.05. The GFI, AGFI, and CFI values all exceeded 0.9. These revised models were adopted as the final ones.

In the preschoolers (Figure 2), the total sleeping habits score, total life skills score, and total healthy diet score had a direct effect on the total physical activity score and the total PPH score. The total physical activity score had a direct effect on the total PPH score. In the elementary school students (Figure 3), age, sex, total sleeping habits score, total life skills score, and total healthy diet score had a direct effect on the total physical activity score and total PPH score, respectively. The total physical activity score was seen to have a direct effect on the total PPH score. Both the preschoolers’ revised model (Figure 2) and the elementary school children’s revised model (Figure 3) showed a significant indirect effect of sleep habits, life skills, and healthy diet on PPH via physical activity.

## 4. Discussion

This study examined the pathways by which lifestyle factors are associated with PPH in preschoolers and elementary school children during the COVID-19 pandemic. Although prior research suggested that lifestyle factors were significant, they failed to show the associated processes or the relative magnitude of the effects of these factors. Furthermore, it is still unknown how these lifestyle factors relate to mental health in children during the COVID-19 period. This study obtained the following findings. First, “life skills” were the factor most strongly associated with PPH in both preschoolers and elementary school children. Additionally, it was discovered that “life skills” had a positive correlation with PPH via physical activity and were associated with the physical activity score. The life skills examined in the current study included independently carrying out daily tasks, such as preparing for the following day, cleaning, eating, and dressing and undressing. Children who have acquired “life skills” such as these are more likely than those who have not to take on challenges without giving up easily during the stages of development for preschoolers and in the first grade of elementary school [21]. Cleaning the house has been found to be an effective coping strategy during the COVID-19 pandemic and it has been shown to increase resilience in the face of adversity [22]. Life skills are effective at enhancing mental health, according to earlier studies conducted with adults. 

The current study is the first to show that, during the COVID-19 pandemic, “life skills” are both directly and indirectly related to PPH in both preschoolers and elementary school children. The maintenance of children’s mental health during the pandemic depends on routines that include elements such as play, reading, rest, and physical activity and that are consistent with pre-pandemic routines [23], with school playing a key role in these routines [24]. A common recommendation provided by mental health professionals was to stay active, even while being isolated inside the house [18]. Being active through life activities, such as tidying up one’s surroundings, may be useful for one’s mental health. Furthermore, exercise and play require preparation and cleanup. Teaching exercise habits and life skills simultaneously may make it easier to improve children’s behavior. The practical significance of these findings is that they point to the significance of adults interacting with children during the prolonged effects of the COVID-19 pandemic in a way that encourages them to exercise autonomy in caring for daily tasks at home and at school, such as preparing for the next day, cleaning up, eating, and dressing and undressing themselves. 

Second, with complaints of stomachaches, nausea, daytime fatigue, poor sleep quality, and irritability, the PPH examined in the current study suggests that children feel unwell and stay home from school or go to the nurse’s office [25]. Age and sex were not related to PPH and “physical activity” in the preschool age group in this study, but they had a negative impact on PPH and “physical activity” in the elementary school age group. Thus, we discovered that, as elementary school children get older, girls’ mental health is worse than that of boys. In a previous study, high school students were more likely than junior high school students to experience symptoms of anxiety and depression, with girls showing particularly strong tendencies [2]. Children in elementary school have demonstrated difficulty in emotional regulation and an increase in hyperactivity and inattentiveness during the COVID-19 pandemic [26], and it has been noted that the proportion of children who are irritable is roughly 10 points higher in the last two years of elementary school than in the first two years [27]. According to the results of the current study, efforts to maintain mental health that take sex and age into account are required not only for junior high and high school students, but also for elementary school children.

Third, among elementary school children, age demonstrated the greatest negative effect on physical activity of all factors. In other words, it was found that the older the elementary school children, the lesser they exercised during the pandemic. Furthermore, this trend was more pronounced in girls than in boys. The physical activity habits examined in the present study comprised active movements such as playing outside, using exercise and playground equipment, parents and children engaging in physical exercise together, and exercising to the point of sweating. Amidst the decline in physical activity during the COVID-19 pandemic, children who played outside (such as playing in the yard, riding a bicycle, going for walks with their families, etc.) reported to have experienced an improved mood [28]. Mentally healthy young people can enter adulthood with happiness and confidence and are able to cope with adversities [29]. Contrastingly, compared with physically active adult women, women who engaged in little physical activity during the COVID-19 pandemic reported to have lower social, emotional, and psychological well-being and significantly higher anxiety, while those who engaged in exercise, even if lower in intensity than that by men, experienced improved emotional well-being [30]. Stress symptoms associated with the COVID-19 pandemic were especially pronounced in women, while a low level of physical activity has been observed among women in various age groups [1,30]. The current study also suggests that elementary school girls need encouragement to be physically active, and it highlights the need to give them support and opportunities that would promote physical activity, such as conducting physical activities that parents and kids can do together.

In conclusion, this study revealed the relationship between sleep, life skills, and healthy eating and PPH directly or indirectly via exercise habits in children during the prolonged COVID-19 pandemic in Japan. 

### Limitations

The limitations of the present study are as follows. 

The present study dealt only with lifestyle factors for which behavior can be changed through interventions at home and school. Consequently, only a small number of items were included, and other factors associated with children’s physical activity habits and PPH may have been omitted.This study is a cross-sectional study, and the data obtained were not assessed in terms of causality.An analysis utilizing a structural equation model with latent variables needs to be conducted to better determine how changes in children’s lifestyle factors and physical activity affect PPH.

## 5. Conclusions

The current study addressed latent factors connected to children’s exercise habits and PPH that can be improved through interventions at home or school. Therefore, this study is distinguished by the findings’ potential for practical application, which may help in modifying children’s behavior. The findings of the present study are summarized as follows. A path analysis of preschoolers and elementary school children suggested that learning life skills can improve PPH, even during the COVID-19 pandemic. Moreover, mental health was found to decline among elementary school students as they aged, and girls were found to have worse mental health than boys. The findings of the present study may be useful in considering interventions at home and school for improving the mental health of children during the COVID-19 pandemic. 

## Figures and Tables

**Figure 1 children-10-00943-f001:**
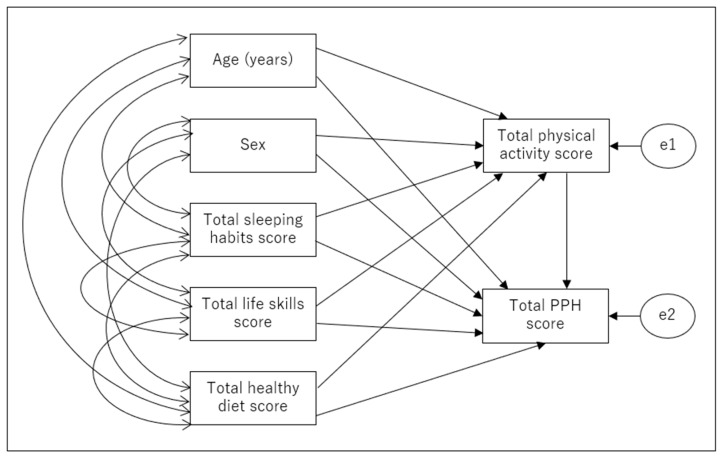
Path diagram of the initial theoretical model. PPH: physical/psychosocial health.

**Figure 2 children-10-00943-f002:**
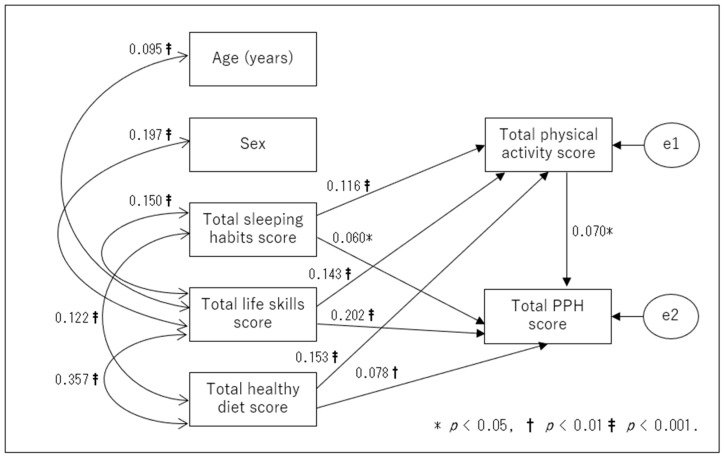
Revised model (preschool children). PPH: physical/psychosocial health.

**Figure 3 children-10-00943-f003:**
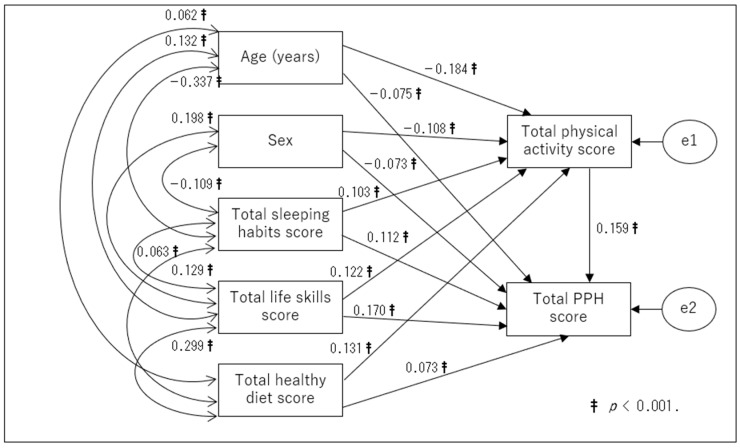
Revised model (elementary school student). PPH: physical/psychosocial health.

**Table 1 children-10-00943-t001:** Descriptive statistics of variables included in the structural equation model by school.

	Pre-Schoolers (*n* = 1183)			Elementary School Children (*n* = 3156)		
	Mean (SD)	Range	Cronbach’s α	Mean (SD)	Range	Cronbach’s α
1. Age (in years)	4.00 (0.80)	3–5		8.39 (1.68)	6–11	
2. Sex (male: 1 and female: 2)	1.49 (0.50)	1–2		1.50 (0.50)	1–2	
3. Total sleeping habits score	10.16 (2.09)	4–16	0.78	9.76 (2.07)	4–16	0.75
4. Total physical/psychosocial health score	22.16 (3.53)	8–28	0.71	21.43 (3.88)	7–28	0.72
5. Total physical activity score	12.07 (2.30)	4–16	0.76	11.38 (2.78)	4–16	0.78
6. Total life skills score	10.87 (2.61)	4–16	0.74	11.93 (2.69)	4–16	0.78
7. Total healthy diet score	5.12 (1.58)	2–8	0.77	5.91 (1.57)	2–8	0.71

Note: Proportions of male and female preschoolers and elementary school children were 50.9% and 49.1% and 49.9% and 50.1%, respectively.

**Table 2 children-10-00943-t002:** Correlation coefficients of variables in the structural equation model by school.

	1	2	3	4	5	6	7
1. Age (in years)		−0.03	−0.01	0.04	0.05	0.107 ‡	0.05
2. Sex (male: 1 and female: 2)	−0.01		−0.04	0.00	0.02	0.198 ‡	0.02
3. Total sleeping habits score	−0.337 ‡	−0.100 ‡		0.109 ‡	0.155 ‡	0.140 ‡	0.122 ‡
4. Total PPH score	−0.117 ‡	−0.060 ‡	0.197 ‡		0.140 ‡	0.255 ‡	0.175 ‡
5. Total physical activity score	−0.193 ‡	−0.086 ‡	0.199 ‡	0.235 ‡		0.216 ‡	0.220 ‡
6. Total life skills score	0.130 ‡	0.207 ‡	0.063 ‡	0.194 ‡	0.123 ‡		0.367 ‡
7. Total healthy diet score	0.062 ‡	0.035 ‡	0.126 ‡	0.159 ‡	0.166 ‡	0.306 ‡	

Note: Data are Pearson correlation coefficients. Upper diagonal: preschoolers (*n* = 1183) and lower diagonal: elementary school students (*n* = 3156). PPH: physical/psychosocial health. ‡ *p* < 0.001.

**Table 3 children-10-00943-t003:** Goodness of fit indices for structural equation models.

Model	Χ^2^	df	*p*	GFI	AGFI	CFI	RMSEA
Age 3–5 years, *n* = 1183							
Initial model	0.768	1.000	0.381	1.000	0.995	1.000	0.000
Revised model (Figure 2)	11.896	9.000	0.219	0.997	0.991	0.994	0.017
Age 6–11 years, *n* = 3156							
Initial model	0.366	1.000	0.545	1.000	0.999	1.000	0.000
Revised model (Figure 3)	4.423	2.000	0.110	1.000	0.994	0.999	0.020

Note: GFI: goodness of fit index; AGFI: adjusted GFI; CFI: comparative fit index; RMSEA: root-mean-square error of approximation.

## Data Availability

The data we used to derive our findings are unsuitable for public deposition due to ethical restrictions and specific legal frameworks in Japan. It is prohibited by the Act on the Protection of Personal Information (Act No. 57 of 30 May 2003, amended on 9 September 2015) to publicly deposit data containing personal information. The Ethical Guidelines for Epidemiological Research enforced by the Japan Ministry of Education, Culture, Sports, Science, and Technology and the Ministry of Health, Labor and Welfare also restrict the open sharing of the epidemiologic data.

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
