# Peer review of "Lifestyle Pathways Affecting Children’s Mental Health in Japan during the COVID-19 Pandemic"

_children, 2023, doi:10.3390/children10060943_

Round 1

Reviewer 1 Report

The authors undertook to implement an interesting topic on Lifestyle pathways affecting children's mental health in Japan during the COVID-19 pandemic. The study was well planned and conducted. Introduction, purpose, material and methods are described in detail. The description of the results also raises no objections. The conclusions are clear. In preparing the manuscript, they used the current literature.

Author Response

Thank for your positive feedback.

In response to other Reviewers comment, we improved the paper.

Changes to the manuscript in response to Reviewer 2 are shown in green font. For Reviewer 3 we used pink font, and the blue font was used for comments common to Reviewer 2 and 3. Finally, the manuscript has been edited by a native speaker of English, who is also a professional academic editor. Corrections in the revised manuscript are shown in orange font.

Reviewer 2 Report

Reviewer comments about the study:

Lifestyle pathways affecting children’s mental health in Japan during the COVID-19 pandemic

1-Study aim: we conducted a path analysis and looked at the individual pathways by which lifestyle factors affect the mental health of preschoolers and elementary school children.

2-Reviewer comment: According my knowledge the study aim is confuse. The authors explain the statistical method of study; however, the write form does not reach a level of understand for reader. It is hard to find the perspective of authors. I suggest that the study aim better starts with a direct verb sentence example: “the present study aim is to describe, or to identify, or to verify the relationship between lifestyle factors and mental health of preschoolers…”

3-Reviewer comment about the models (original/revised)

A)      Why the authors fit a correlation between variables and errors?

B)      Considering the presented structural equation, the authors considering that physical activity is a mediator factor to stablish relations among lifestyle and mental health. Would be possible that authors presenting the indirect relations intermediate by physical activity. It is important to understand if physical activity practice affect the other health aspects. But, whether not the intention of authors to consider this a mediator factor a suggest that authors change the model, because physical activity could be just exogenous factor and not an endogenous mediator.

4-General comment: Analyzing a structural equation model is always a challenge, because it depends on the hypotheses that permeate the authors' ideas about the research topic. Even so, my general suggestions for the article are related to how the model is conceived. Considering that mind and lifestyle are variables that cannot be disconnected, I think that a structural equation model with latent variables for each questionnaire, taking into account all numerical responses and designed as a non-recursive model, of cyclical relationships, could improve the goodness of fit, and make it clearer if the theoretical matrix fits with the tested database. Another suggestion is that the authors present and base themselves on previous models, to theoretically justify the proposal of a structural equation model. I suggest reading structural and complex analysis of pathways, and if authors to consider adequate include some articles about it in present theoretical references:

https://doi.org/10.1080/02568543.2019.1577775

https://doi.org/10.1123/jsep.32.1.99

https://doi.org/10.5993/AJHB.43.1.6

https://doi.org/10.1016/j.mhpa.2013.06.002

https://doi.org/10.1016/j.ijpam.2020.11.004

https://doi.org/10.3390/nu11081953

https://doi.org/10.3389/fped.2021.656916

Author Response

We wish to express our appreciation to the reviewers for their useful comments, which helped us to significantly improve the paper. Changes to the manuscript in response to the comments by Reviewer 2 are shown in green-colored font. For the comments by Reviewer 3 we used pink-colored font, and the blue-colored font was used for the comments common to Reviewers 2 and 3. Finally, the manuscript has been edited by a native speaker of English, who is also a professional academic editor. Other corrections in the revised manuscript are shown in orange-colored font.

  1. Study aim: we conducted a path analysis and looked at the individual pathways by which lifestyle factors affect the mental health of preschoolers and elementary school children.
  2. According my knowledge the study aim is confuse. The authors explain the statistical method of study; however, the write form does not reach a level of understand for reader. It is hard to find the perspective of authors. I suggest that the study aim better starts with a direct verb sentence example: “the present study aim is to describe, or to identify, or to verify the relationship between lifestyle factors and mental health of preschoolers…”

Response 1 and 2: Thank you for your comment. We have changed the following text (lines 52-54):

(before)

we conducted a path analysis and looked at the individual pathways by which lifestyle factors affect the mental health of preschoolers and elementary school children.

(after)

the present study’s aim is to describe the relationship between lifestyle factors and mental health of preschoolers and elementary school children during the COVID-19 pandemic by gender and age.

3.Reviewer comment about the models (original/revised)

A) Why the authors fit a correlation between variables and errors?

Response 3A: Thank you for your comment. We have now removed errors for unaffected variables (“Total sleeping habits score,”“Total life skills score,” “Total healthy diet score”). We have performed a path analysis with the corrected model and obtained the same results. Thus, we revised the figures (Figs 1, 2 and 3).

B) Considering the presented structural equation, the authors considering that physical activity is a mediator factor to stablish relations among lifestyle and mental health. Would be possible that authors presenting the indirect relations intermediate by physical activity. It is important to understand if physical activity practice affect the other health aspects. But, whether not the intention of authors to consider this a mediator factor a suggest that authors change the model, because physical activity could be just exogenous factor and not an endogenous mediator.

Response 3B: Thank you for your constructive feedback. We have read the suggested literature and added relevant references that could enforce our theoretical model (reference numbers 15, 17, 18, and 19) and articulated our theoretical model. We have added the following text (lines 95-109, lines 145-147, lines 149-150, lines 161-163). Specifically, as for your suggestion to regard mental health as an exogenous variable leading to physical exercise, we have added an explanation about our theoretical model in the manuscript.

4. General comment: Analyzing a structural equation model is always a challenge, because it depends on the hypotheses that permeate the authors' ideas about the research topic. Even so, my general suggestions for the article are related to how the model is conceived. Considering that mind and lifestyle are variables that cannot be disconnected, I think that a structural equation model with latent variables for each questionnaire, taking into account all numerical responses and designed as a non-recursive model, of cyclical relationships, could improve the goodness of fit, and make it clearer if the theoretical matrix fits with the tested database. Another suggestion is that the authors present and base themselves on previous models, to theoretically justify the proposal of a structural equation model. I suggest reading structural and complex analysis of pathways, and if authors to consider adequate include some articles about it in present theoretical references:

Response 4: Thank you for your comment. We appreciate that you introduced us to the previous studies. Following the comments, we theoretically reinforced the hypothetical model by adding relevant references (15,17) from the suggested literature. Furthermore, the model fitness indices were very good. Thus, we did not perform the analysis in the study. However, we referred to this as a limitation to the study (lines 252-254).

Reviewer 3 Report

Thanks for giving me the opportunity to review this paper.
I have several concerns regarding this paper which are given below:

1. The contribution of your study is missing in the introduction section. Explain it at the end of the introduction.
2. What has been done previously and what do you want to do in this study? The problem statement and research gap are missing in the introduction section.
3. Explain your methodology part a bit more.
4. Include your practical and theoretical contributions and link them to the previous research and discuss them thoroughly.
5. Discussion and conclusion seem a summary. It is highly recommended to link your findings with previous studies. Also, discuss the contribution of your study such as what has been done and what you want to do and what is your novelty.

Good luck    

n/a

Author Response

We wish to express our appreciation to the reviewers for their useful comments that helped to significantly improve the manuscript. Changes to the manuscript in response to the comments by Reviewer 3 are shown in pink-colored font. For the comments by Reviewer 2, we used green-colored font, and the blue-colored font was used for the comments common to Reviewers 2 and 3. Finally, the manuscript has been edited by a native speaker of English, who is also a professional academic editor. Other corrections in the revised manuscript are shown in orange-colored font.

  1. The contribution of your study is missing in the introduction section. Explain it at the end of the introduction.

Response 1: Thank you for your constructive feedback. Following the comment, we have added the contribution of our study at the end of Introduction (lines 46–54).

  1. What has been done previously and what do you want to do in this study? The problem statement and research gap are missing in the introduction section.

Response 2: Thank you for your comment. We have added a description elaborating on this information in the manuscript. (lines 38–42)

  1. Explain your methodology part a bit more.

Response 3:  Thank you for your comment. We have added references (reference number 15,17,18,19) and articulated our theoretical model more thoroughly in the revised manuscript (lines 95-109).

  1. Include your practical and theoretical contributions and link them to the previous research and discuss them thoroughly.
  2. Discussion and conclusion seem a summary. It is highly recommended to link your findings with previous studies. Also, discuss the contribution of your study such as what has been done and what you want to do and what is your novelty.

Responses 4 and 5: Thank you for your comment. We have revised Discussion and Conclusion so that practical and theoretical contributions of the study in the linkage of the previous research are clearly presented (lines 174- 266, specifically lines 174-178, lines 195-199, lines 219-222, lines 240-242, lines 257-260). Moreover, we also added sentences explaining the novelty of the study (lines 260-266).
